# Comparing the number of outdoor sugar-sweetened beverage and caffeinated beverage advertisements near schools by school type and school-level economic advantage

**Phoebe R. Ruggles**[1], **Keryn E. Pasch**[1]\*, **Natalie S. Poulos**[2], **Jacob E. Thomas**[1]

**1** Department of Kinesiology & Health Education, University of Texas at Austin, Austin, Texas, United States of America, **2** Department of Nutritional Sciences, The University of Texas at Austin, Austin, Texas, United States of America

\* kpasch@austin.utexas.edu

**Data Availability Statement:** All excel files are available from the Texas Data Repository on the

## Abstract

### Introduction

Sugar-sweetened beverage and caffeinated beverage consumption are associated with a variety of health issues among youth. Food and beverage marketing has been shown to affect youth's preferences, purchases, and consumption of marketed products. Previous research suggests that outdoor food and beverage marketing differs by community demographics, with more advertisements in lower-income communities and near schools. The purpose of this study is to examine the density of sugar-sweetened and caffeinated beverage advertisements near schools by school type (middle vs. high school) and by school-level SES.

### Methods

Data are from the Outdoor Measuring and Evaluating the Determinants and Influence of Advertising (MEDIA)study, which documented and described all outdoor food and beverage advertisements near 47 middle and high schools in 2012. Beverage advertisements were categorized as: sugar-sweetened/caffeinated, sugar-sweetened/non-caffeinated, non-sugar-sweetened/caffeinated, or non-sugar-sweetened/non-caffeinated. Schools were categorized by type (middle vs high) and by SES as determined by the percentage of students qualifying for free or reduced-price lunch. Bootstrapped non-parametric Mann-Whitney U tests compared the number of advertisements in each category by school type and school-level SES (higher vs lower).

### Results

Compared to schools with higher SES, schools with lower SES had significantly more advertisements for sugar-sweetened/non-caffeinated beverages (Median$_{low}$ = 28.5 (IQR 17–69),

University of Texas at Austin Dataverse Collection database doi:10.18738/T8/ELJLVA.

**Funding:** Recipient: KEP This work was supported by the National Institutes of Health National Cancer Institute Grant #R03CA158962. https://www.nih.gov/about-nih/what-we-do/nih-almanac/national-cancer-institute-nci The funders had no role in study design, data collection and analysis, decision to publish, or preparation of the manuscript.

**Competing interests:** The authors have declared that no competing interests exist.

vs Median$_{high}$ = 10.5 (IQR 4–17) (p = 0.002)., sugar-sweetened non-caffeinated (Median$_{low}$ = 46 (IQR 16–99) vs Median$_{high}$ = 13.5 (IQR 6–25), p = 0.002), -sugar-sweetened caffeinated (Median$_{low}$ = 12 (IQR 8–19) vs Median$_{high}$ = 6 (IQR 2–8), p = 0.000), and non-sugar-sweetened non-caffeinated (Median$_{low}$ = 30 (IQR 13–65) vs Median$_{high}$ = 14 (IQR 4–29), p = 0.045).There were no significant differences by school type.

## Conclusion

This study adds to the literature demonstrating pervasive marketing of unhealthy products in lower-income communities. Disproportionate exposure to sugar-sweetened and caffeinated beverage advertisements in lower-income communities may contribute to the disparities in associated health outcomes by economic status.

## 1. Introduction

Reducing sugar-sweetened beverage (SSB) consumption has been the target of many attempts to combat the national obesity crisis, particularly in children. As of 2016, SSBs accounted for 24% of all added sugars consumed by people ages two and up in the United States [1]. While SSB consumption among children has declined in recent years [2–4], children still consume SSBs regularly. In 2018, 72–77% of youth ages 2–19 (varying by demographic characteristics) reported consuming at least one SSB in the past week [2]. Regular consumption of SSBs increases total energy intake and has been associated with increased weight, Type II Diabetes, and metabolic syndrome [5, 6], as well as behavioral and mental health problems [7].

While SSB consumption among children is a primary public health concern, caffeinated beverages also pose issues to children's health.[8] Caffeine is a widely used psychostimulant which acts on both the central and peripheral nervous systems, and is commonly consumed by children and adolescents [9]. It is estimated between 71% and 75% of children ages two to 19 consume caffeine on any given day [9–11]. The prevalence of caffeine consumption among children and adolescents generally increase with age, with approximately 75.3% of youth ages 12–16 consuming caffeine daily, rising modestly to 75.8% of youth ages 17–18 [10]. Additionally, the average amount of caffeine intake increases as children age, with 6–11 year olds consuming approximately 25 mg/day and children ages 12–17 consuming an average of 50 mg/day [9]. Like SSBs, caffeine use is associated with a variety of physical health, emotional health, and social/behavioral conditions in children and adolescents. For example, poor sleep, fatigue, headache, nausea, depressive symptoms, anxiety, and feelings of stress are all associated with caffeine consumption [9, 12–14] Additionally, caffeine is associated with children and adolescent risk-taking behaviors, including drug and alcohol use, as well as feelings of anger and violence [9]. Caffeine is also associated with a higher risk of developing obesity in children [15], likely due to its inclusion in sugary drinks [12]. Soft drinks are the most common source of caffeine consumed by children and adolescents, though coffee and energy drink consumption is rising among adolescents [10, 16]. Additionally, caffeinated SSBs are more habitually consumed and better liked by adolescents as compared to non-caffeinated beverages [17]. While the Dietary Guidelines for Americans suggests that adults over the age of 18 can safely consume up to 400 mg of caffeine per day [1], the American Academy of Pediatrics and the USDA discourage caffeine consumption in children and adolescents [1, 18].

Both sugar-sweetened and caffeinated beverages are heavily marketed, with over $1 billion spent on SSB and energy drink advertising in 2018 [19]. The effects of persuasive food and

beverage marketing on youth are well documented [20–22]. Numerous studies have demonstrated the instantaneous effects of viewing branded food and beverage product advertisements on children's short-term preferences for and attitudes toward advertised products [23, 24]. as well as their consumption of advertised products [22, 23, 25]. Research suggests that between 42% and 54% of middle and high school-aged adolescents (ages 12–17) report seeing or hearing at least one beverage advertisement per day [26]. Moreover, advertising susceptibility in youth is a reliable predictor of SSB product preferences [27] and SSB consumption [28].

The advertising environment near schools is critical to understanding children's advertising exposure, given the extensive time children spend at school. Advertising exposure is associated with consumption behaviors in children [22], and because children travel to and from school daily, they are frequently exposed to advertisements near schools. The quality and density of food outlets near schools tends to vary by neighborhood income, such that lower income schools tend to have a greater number of total food outlets as well as fast-food outlets within walking distance, compared to schools in higher-income areas [29]. Most students attend schools within their assigned districts, based on the location of their residence [30].

The saturation of food and beverage advertising likely varies by school type and SES. While differences in the prevalence of food and beverage advertising in areas around middle schools compared to high schools have not been specifically examined, other measures that have been used may suggest a difference in SSB and caffeinated beverage marketing. First, the consumption of SSBs and caffeinated beverages vary by age and grade level. For example, middle school students consume more SSBs than high school students [31] but older adolescents consume more caffeine than younger adolescents [10]. Knowing the effects of advertising exposure on food and beverage consumption in youth, it is plausible that the outdoor SSB marketing environment is more saturated near middle schools while the caffeinated beverage marketing environment is more saturated near high schools. Further, economic factors may also play a role in outdoor food and beverage marketing around schools. Research consistently shows disparities in unhealthy food and beverage advertising exposure by SES [32]. Previous research has found that neighborhood-level income demographics are also associated with the outdoor food and beverage advertising environment, with lower-income areas consistently having higher densities of food and beverage marketing than higher-income areas [33–37]. However, few have specifically examined if this economic disparity in advertising applies to school environments [37], nor if the types of beverages advertised vary by neighborhood income level.

Given the effects of both sugar-sweetened and caffeinated beverage consumption on health, the documented history of predatory marketing practices to children, and the disparities observed in diet-related health risks and food and beverage marketing practices, it is crucial to further examine the outdoor beverage advertising environment to youth. Because SSBs are frequently caffeinated, and caffeinated beverages are frequently sugar-sweetened, it is beneficial to examine these two beverage types in tandem when evaluating factors that contribute to their consumption. Thus, the purpose of this study is to examine the densities of outdoor sugar-sweetened and caffeinated beverage advertisements to determine if the density varies by school characteristics (school-level and SES). Given previous studies, we developed the following hypotheses:

1. Middle schools will have greater numbers of SSB advertisements than high schools.

2. High schools will have greater numbers of caffeinated SSB advertisements than middle schools.

3. Schools with ≥60% of students qualifying for free or reduced priced lunch (FRPL) will have greater numbers of SSB advertisements compared to schools with <60% of students qualifying for FRPL.

4. Schools with ≥60% of students qualifying for FRPL will have greater numbers of caffeinated SSB product advertisements compared to schools with <60% of students qualifying for FRPL.

## 2. Methods

Data are from the Outdoor MEDIA (Measuring and Evaluating the Determinants and Influence of Advertising) Study [38, 39]. This study documented the outdoor food and beverage advertising within a one-half mile radius of 34 middle schools, 13 high schools, and nine hospitals in the Austin, Texas area in 2012. All 32 public middle and high schools in the one school district and 15 public middle schools from nearby areas in other school districts were included in this study. The nine hospitals were included as comparators for schools, as they are also community-serving institutions. For this study, data include only those advertisements around middle and high schools. No human subjects were involved in this study; therefore, Institutional Review Board approval was not needed.

### 2.1 Data collection

Prior to data collection, maps of the areas of interest were created using ESRI ArcGIS [40]. An 800-meter radius (approximately one-half mile) buffer was created around each school and hospital. Driving routes were then mapped within these buffers in order to ensure data collectors included every street.

Using a detailed data collection protocol adapted from previous work [41] and a validated data collection tool [39], trained teams of data collectors documented all forms of outdoor marketing materials (e.g., advertisements attached to establishments, free standing advertisements, menus, outlet logos) within one half mile of each school. This data collection tool, powered by FileMaker Pro [42] allowed data collectors to photograph and describe the advertisements on an iPod Touch while in the field and then do additional coding in the lab. Data collectors used the detailed driving directions to systematically drive the ½ mile radius around each school, stopping each time they observed a food or beverage advertisement or establishment. A food/beverage advertisement was defined as any sign promoting food or beverages (including logos and words). The data collectors took pictures of each advertisement, documented basic details about the advertisement, including the category of advertisement (e.g., establishment advertisement, free standing advertisement), type of advertisement (e.g., billboard, convenience store, bus stop) and subject of the advertisement (e.g., food/beverage, tobacco, alcohol) and noted the latitude and longitude. Once data collection was complete, trained research assistants coded the data in the research lab for additional details and content, including brand and product information.

### 2.2 Coding

Inter-rater reliability analyses were conducted to document the reliability of documenting advertisements and coding of the descriptive details of each advertisement. Percent agreements were calculated using data from random sample of seven schools, approximately 15% of the total sample. Percent reliability was high overall for documenting the presence of advertisements (0.71, SE = 0.45) and coding of the content of advertisements, including category, type, and subject (0.84, SE = 0.22). In addition, the mean percent agreement between coders for descriptive coding (which coded the type of products present) was also high at 0.912 (SD = 0.02) [39].

Each advertisement containing an image of a beverage item was coded for "variety" (sugar-sweetened beverage or non-sugar-sweetened beverage), "subject" (e.g., soft drink, tea, coffee),

"brand" (e.g., Coca-Cola, Sonic), and "item" (e.g., Coke Zero, Frappuccino, Red Bull Original). Varieties were considered "sugar-sweetened beverages" if they were sweetened with sugar or sugary product (e.g., corn syrup). "Non-sugar-sweetened beverages" included all beverages that were unsweetened or sweetened with a calorie-free artificial sweetener (e.g., sucralose). All items coded as sugar-sweetened or non-sugar-sweetened beverages were included for analysis in this study. Caffeine content was not included in the initial coding, so all beverage items were re-coded by a single coder (PRR) in 2021 for caffeine content. Items were coded as a *caffeinated beverage* if the subject, brand, or item indicated a caffeinated product (e.g., Coca-Cola Classic, coffee, black tea, Red Bull, energy drink). If the initial coding did not indicate a "subject" or "item" that identified a product that could be caffeinated, it was excluded from caffeinated beverage analyses. For example, if an item was coded as a soft drink under "variety," but there was no "brand" or "item" information, the item would be coded as not caffeinated. All caffeinated soft drinks, tea, coffee, and energy drinks that were identifiable to the item level were coded as caffeinated unless clearly labeled as non-caffeinated or decaffeinated.

Using this information, four categories were created: 1) Sugar-sweetened caffeinated, 2) sugar-sweetened non-caffeinated, 3) non-sugar-sweetened caffeinated, and 4) non-sugar-sweetened non-caffeinated. *Sugar-sweetened caffeinated* beverages include Pepsi, Monster, sweet tea, and other caffeinated beverages sweetened with a caloric sweetener. *Sugar-sweetened non-caffeinated* beverages include Sprite, Gatorade, fruit drinks, and other non-caffeinated beverages sweetened with a caloric sweetener. *Non-sugar-sweetened caffeinated* beverages include unsweetened coffee, Diet Coke, Red Bull Sugar-free, and other caffeinated beverages that are not sweetened or sweetened with a non-caloric sugar substitute. *Non-sugar-sweetened non-caffeinated* beverages include water, plain milk, 100% fruit juice, Diet 7-Up, and other non-caffeinated beverages not sweetened or sweetened with a non-caloric sugar substitute. Many advertisements contained more than one beverage, often including products from multiple categories. To best capture the exact products advertised, each beverage within an advertisement was coded individually up to a total of four beverages.

### 2.3 School-level SES

Schools were dichotomized into higher and lower SES categories using school level Free or Reduced-Price Lunch (FRPL) data from the Texas Education Agency [43]. To qualify for free school lunch, a student's family must be at or below 130% of the federal poverty line, while students qualifying for a reduced-price lunch are at or below 185% of the federal poverty line [44]. Schools were coded as higher SES schools if <60% of students enrolled qualified for FRPL while schools with e60% of students qualifying for FRPL were coded as lower SES schools. Previous research has found that school-level FRPL is an acceptable proxy for individual-level household income [45]. The 60% threshold was chosen because it is the threshold the Food and Nutrition Service uses to determine school-lunch reimbursement rates, where schools with e60% of students qualifying for FRPL receiving slightly higher reimbursements than schools with <60% of students qualifying for FRPL [46]. This threshold has also been used to categorize school-level income in other research [47].

### 2.4 Statistical analyses

The total number of advertised beverages of each category around each school was calculated. Schools where no beverage advertisements were observed were dropped from these analyses (n = 17). There were no meaningful differences in demographics (racial/ethnic composition of the study body or school-level SES) between the included and excluded schools. On inspection, the data did not meet normal distribution assumptions, therefore we used bootstrapped non-

parametric Mann-Whitney U tests with 1,000 repetitions to compare the number of advertisements in each category by school type and school-level SES. Given the relatively small sample of schools (n = 32), we used bootstrapping to ensure better precision in our models [48, 49]. All analyses were conducted using Stata version 17.

School-level racial and ethnic composition was considered as another school level variable, given that previous studies have found differences in advertising by neighborhood racial/ethnic composition [35, 50]. However, at the school-level, racial/ethnic composition and school-level SES were highly colinear and thereby excluded from our analyses.

## 3. Results

A total of 10,070 advertisements were documented around 34 middle schools and 13 high schools in the Austin, TX area. For this study, schools that had no beverage advertisements were dropped from analyses, leaving a total of 32 schools, consisting of 23 middle and nine high schools. Table 1 describes the racial/ethnic composition of the included schools by school type and school-level SES, as well as the proportion of students qualifying for free lunch, reduced-priced lunch, and free or reduced priced lunch by school-level SES.

Of schools that had beverage advertisements, 2,159 advertisements portrayed one beverage, 292 advertisements portrayed two beverages, 214 advertisements portrayed three beverages, and 214 advertisements portrayed four beverages. When broken down by beverage type, there were 1273 images of sugar-sweetened caffeinated beverages, 1577 images of sugar-sweetened non-caffeinated beverages, 343 non-sugar-sweetened caffeinated beverages, and 1048 non-sugar-sweetened non-caffeinated beverages, for a total of 4,214 advertised beverages.

Schools with lower SES had, on average, significantly more beverages of all types compared to schools with greater SES. The median number of sugar-sweetened caffeinated beverages advertised near lower SES schools was 28.5 (IQR 17–69), whereas the median number near higher SES schools was only 10.5 (IQR 4–17, p = 0.002). Similar significant differences by school-level SES were observed for sugar-sweetened non-caffeinated ($Median_{low}$ = 46 (16–99) vs $Median_{high}$ = 13.5 (6–25), p = 0.002), non-sugar-sweetened caffeinated ($Median_{low}$ = 12 (8–19) vs $Median_{high}$ = 6 (2–8), p = 0.000), and non-sugar-sweetened non-caffeinated ($Median_{low}$

**Table 1. Socio-demographic data by school type and SES among middle and high schools in the Austin, Texas area in 2010–2011.**

|  | Middle Schools | High Schools | e60% FRPL | <60% FRPL |
|---|---|---|---|---|
| **Mean # of Students** | 901.95 | 1420.55 | 940.17 | 1186.21 |
| **Race/Ethnicity (%)** |  |  |  |  |
| American Indian/Alaska Native | 0.3 | 0.22 | 0.18 | 0.41 |
| Asian/Pacific Islander | 5.60 | 1.80 | 2.63 | 6.98 |
| Black/African American | 12.49 | 17.22 | 15.91 | 11.14 |
| Hispanic | 52.83 | 54.64 | 70.76 | 30.94 |
| White | 28.85 | 25.79 | 10.54 | 50.43 |
| **FRPL Breakdown (%)** |  |  |  |  |
| Free Lunch | - | - | 80.6 | 32.1 |
| Reduced-Price Lunch | - | - | 9.1 | 6.1 |
| Free or Reduced-Price Lunch | - | - | 89.7 | 38.2 |

Note: Data obtained from the Texas Education Agency. Hispanic ethnicity is not distinguished by race (e.g., Non-Hispanic White, Hispanic Black/African American) in available demographic data.

= 30 (13–65) vs Median$_{high}$ = 14 (4–29), p = 0.045). Contrary to our hypotheses, there were no significant differences by school type (Table 2).

## 4. Discussion

In this study, we examined the number of outdoor sugar-sweetened and caffeinated beverages advertised within a half-mile radius of a sample of middle and high schools in central Texas. The areas near schools where ≥60% of students qualified for FRPL had advertisements containing significantly more beverages of all types compared to the areas near schools where <60% of students qualified for FRPL. There were no differences in number of advertisements by school type; middle schools and high schools did not differ statistically in the types of beverages advertised within half-mile of the schools.

Our study is the first to examine how the density of advertisements for caffeinated beverages may vary by school level SES. Few studies take economic status into account when examining caffeine consumption patterns and of these, none, to our knowledge, have found statistically significant variation by income or socioeconomic status [51]. The findings of our study indicate that the density of outdoor beverage advertisements near schools, particularly for sugar-sweetened/non-caffeinated and non-sugar-sweetened/non-caffeinated beverages, significantly differs by school-level SES, but does not differ by school type. This finding concurs with previous work examining the densities of outdoor food and beverage advertisements by economic demographics [35, 52]. Thus, the present study contributes to a growing body of literature suggesting that lower-income areas may be disproportionally saturated with food and beverage advertising as compared to higher-income areas.

The significant differences by school-level SES in the density of beverage advertising may further our understanding of the drivers behind the socioeconomic differences in SSB consumption. For example, because of the effects food and beverage marketing exposure have on children's dietary preferences and consumption habits [53], the disparities in marketing exposure may contribute to disparities in SSB consumption. In turn, socioeconomic differences in SSB consumption may then contribute to the socioeconomic disparities in the multitude of adverse health outcomes associated with SSBs, including obesity [54], and Type II diabetes [55]. Further, economic disparities in the health outcomes associated with caffeine consumption in youth have been observed, including depressed mood and anxiety [56], obesity [54], and poor sleep [57].

Although previous research suggests that middle school-aged children tend to consume more SSBs than high school-aged students [31] and that high school-aged students tend to consume more caffeine than middle school-aged students [10], our study did not find differences in the density of SSB advertising by school type. Our null findings could suggest that marketers view the areas around middle school and high schools as similar and do not differentially target by school type. However, our null findings may also be due to the small sample sizes of each of the school types, particularly for high schools. Therefore, future research in this area is needed with a larger sample of middle and high schools or more targeted school environments to best understand how beverages are advertised around schools.

Because of the health issues associated with regular consumption of unhealthy, frequently advertised products, many public health advocates have called for tighter restrictions on food and beverage marketing to children, citing it as a major factor in the childhood obesity crisis in the United States [58]. Industry self-regulation efforts, such as the Children's Food and Beverage Advertising Initiative (CFBAI) have generally been considered ineffective [59], and do not include regulations for outdoor advertising. Further, CFBAI initiatives only regulate advertisements targeted towards children ages 12 and under, thus leaving most middle and high

**Table 2. Differences in number of beverage advertisements by school type and SES and by beverage category around schools in two central texas counties.**

| Advertisement Type | School Type | | | | | | | Free or Reduced-Price Lunch Eligibility | | | | | | |
|---|---|---|---|---|---|---|---|---|---|---|---|---|---|---|
| | Middle School (n = 23) | | High School (n = 9) | | | | | Lower SES (n = 18) | | Higher SES (n = 14) | | | | |
| | Median (IQR) | Mean (SD) | Median (IQR) | Mean (SD) | Bootstrapped Z-Statistic | Bootstrapped Std. Error | p | Median (IQR) | Mean (SD) | Median (IQR) | Mean (SD) | Bootstrapped Z-Statistic | Bootstrapped Std. Error | p |
| Sugar-Sweetened/ Caffeinated | 17 (9–32) | 26.61 (32.22) | 55 (14–82) | 73.44 (85.04) | 1.57 | 0.96 | 0.10 | 28.5 (17–69) | 53.94 (63.77) | 10.5 (4–17) | 21.57 (36.69) | -3.08 | 0.86 | 0.002 |
| Sugar-Sweetened/ Non-Caffeinated | 25 (8–47) | 50.56 (73.18) | 18 (14–58) | 46 (45.66) | 0.57 | 0.92 | 0.57 | 46 (16–99) | 68.05 (73.08) | 13.5 (6–25) | 25.14 (47.5) | -3.15 | 0.84 | 0.002 |
| Non-Sugar-Sweetened/ Caffeinated | 8 (2–14) | 10.74 (10.51) | 8 (6–16) | 10.66 (6.3) | 0.62 | 0.95 | 0.54 | 12 (8–19) | 14.72 (10.41) | 6 (2–8) | 5.57 (4.35) | -3.64 | 7.84 | 0.00 |
| Non-Sugar-Sweetened/ Non-Caffeinated | 22 (4–61) | 33.78 (34.37) | 28 (13–31) | 30.11 (29.26) | -0.02 | 0.92 | 0.98 | 30 (13–65) | 41.94 (37.5) | 14 (4–29) | 20.93 (20.65) | -2.00 | 0.90 | 0.045 |

Note:

Lower SES indicates schools with ≥60% of students qualifying for Free or Reduced-Price Lunch

Higher SES indicates schools with <60% of students qualifying for Free or Reduced-Price Lunch

school-aged youth unprotected. While an overhaul of food and beverage advertising regulation at the federal level is unlikely, some state and local governments have considered or implemented policies to limit food and beverage marketing to youth [60]. Our findings suggest that food and beverage advertising regulations should expand beyond television to also include the outdoor environment around schools and include protections for older youth. Regulating the outdoor marketing environment to limit children's exposure to unhealthy product marketing, including marketing vehicles that are not explicit but expose youth to product specific advertisements (e.g., vending machines with brand logos) would help to protect youth from extensive advertising exposure. Further, protections should be implemented to reduce targeted advertising in areas with lower SES.

There are several notable strengths in this study. First, this study used a validated tool to objectively document advertisements and used a reliable coding system to advertisements from photos which allowed for detailed documentation of the advertised products. Second, this study examined four separate categories of beverage products. Each of these categories provides a more detailed picture of the types of products that are seen by youth. Finally, this study is the first to our knowledge to examine advertising of caffeinated beverages as a distinct category. This will contribute to further studies on the effects of caffeinated product marketing and the consumption of caffeine in youth.

However, there are also limitations to this study's findings. First, the findings are limited to the two counties in Central Texas, which may limit generalizability. Further, analyses did not consider the density of convenience stores or fast-food establishments within the data collection zones of each school. Because many outdoor food and beverage advertisements appear on the outside of establishments, this may explain some of the variances in advertisement counts. However, if the difference observed between lower and higher SES schools was driven by the number of establishments, the implications for health would remain. Future research should consider measuring distance/proximity to major intersections or business areas, walkability score for the neighborhood, or housing density to assess if other neighborhood characteristics are associated with the number of advertisements. Additionally, the age of the data (collected in 2012) presents limitations, as the marketing environment may have changed over time. However, there have been no systemic changes to the outdoor beverage marketing environment near schools in Austin since data collection. Finally, the small sample size of schools, particularly high schools, limits the power of the analyses, though our bootstrapped analyses improved our ability to detect differences. Future studies with larger sample sizes are needed to continue to explore these associations.

In sum, this study contributes novel insights into the beverage marketing environment near middle and high schools by addressing two specific categories of beverage items that are of particular interest to public health: sugar-sweetened beverages and caffeinated beverages. Previous research has examined both food and beverage advertisements, but this is the first to highlight these specific beverage categories. Given the public health problems linked to SSBs and caffeine consumption in youth, these findings provide specific insight into the environmental contributors to their consumption. This study also adds to a growing body of evidence of discriminatory advertising practices of unhealthy products in lower-income communities.

## Author Contributions

**Conceptualization:** Phoebe R. Ruggles, Keryn E. Pasch, Natalie S. Poulos.

**Data curation:** Keryn E. Pasch, Natalie S. Poulos, Jacob E. Thomas.

**Formal analysis:** Phoebe R. Ruggles, Jacob E. Thomas.

**Funding acquisition:** Keryn E. Pasch, Natalie S. Poulos.

**Investigation:** Natalie S. Poulos.

**Methodology:** Keryn E. Pasch, Natalie S. Poulos.

**Project administration:** Keryn E. Pasch, Natalie S. Poulos.

**Resources:** Keryn E. Pasch, Natalie S. Poulos.

**Supervision:** Keryn E. Pasch, Natalie S. Poulos.

**Writing – original draft:** Phoebe R. Ruggles.

**Writing – review & editing:** Phoebe R. Ruggles, Keryn E. Pasch, Natalie S. Poulos, Jacob E. Thomas.

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
