## [Decision Letter · Decision Letter 0]

7 Aug 2023

PONE-D-23-10045Comparing the Density of Outdoor Sugar-Sweetened Beverage and Caffeinated Beverage Advertisements Near Schools by School Type and School-Level Economic AdvantagePLOS ONE

Dear Dr. Pasch,

Thank you for submitting your manuscript to PLOS ONE. After careful consideration, we feel that it has merit but does not fully meet PLOS ONE’s publication criteria as it currently stands. Therefore, we invite you to submit a revised version of the manuscript that addresses the points raised during the review process.

We look forward to receiving your revised manuscript.

Kind regards,

Habiba I. Ali, PhD, RD, CDE

Academic Editor

PLOS ONE

Journal Requirements:

2. Please note that in order to use the direct billing option the corresponding author must be affiliated with the chosen institute. Please either amend your manuscript to change the affiliation or corresponding author, or email us at plosone@plos.org with a request to remove this option.

4. We note you have included a table to which you do not refer in the text of your manuscript. Please ensure that you refer to Table 1 in your text; if accepted, production will need this reference to link the reader to the Table.

Additional Editor Comments:

Dear authors,

Please fully address all the concerns raised by the reviewers. These concerns are related to but not limited to the presentation of the statistical analysis in the Results section and the need for greater description of the Introduction and the methods sections.

Reviewers' comments:

Reviewer's Responses to Questions

**Comments to the Author**

1. Is the manuscript technically sound, and do the data support the conclusions?

Reviewer #1: Yes

Reviewer #2: Yes

2. Has the statistical analysis been performed appropriately and rigorously? 

Reviewer #1: Yes

Reviewer #2: Yes

3. Have the authors made all data underlying the findings in their manuscript fully available?

Reviewer #1: Yes

Reviewer #2: Yes

4. Is the manuscript presented in an intelligible fashion and written in standard English?

Reviewer #1: Yes

Reviewer #2: Yes

5. Review Comments to the Author

Reviewer #1: Overall, this is a sound study and important to the field of predatory marketing of unhealthy food products. I hope you will give serious consideration to the following comments:

In the results section (and in abstract), I’d be careful of the word “significantly.” We don’t want to confuse statistical significance w. the effect size. I suppose it’s fine as it is but perhaps re-phrase to something like “schools with lower SES had a dramatically higher number of advertisements…” or something like that.

You say: ““Schools with lower SES had more sugar-sweetened/caffeinated and non-sugar-sweetened/non-caffeinated beverage advertisements, though the differences were not statistically significant.” This is not a fair statement. If it’s not statistically significant, it really shouldn’t be mentioned at all. A more academically rigorous statement would be to say: “There were no statistically significant differences between schools with lower SES and higher SES in terms of sugar-sweetened/caffeinated and non-sugar-sweetened/non-caffeinated beverage advertisements.” The results are the results (and I agree that .05 p values are arbitrary). But strong papers do not emphasize (and many don’t even mention) non-significant results. Your study is sound… don’t grasp for straws when you have some perfectly solid results to report.

The results section does seem somewhat scant. Aren’t there more interesting results that can be reported here?

In terms of recommendations: You might consider going beyond recommending that “that food and beverage advertising regulations should expand beyond television to include the outdoor environment around schools / protections for older youth. Remember that soda machines are themselves a marketing vehicle. So you might also mention limiting where these machines are placed.

In the limitations you mentioned that you didn’t consider the density of convenience stores or fast-food establishments, but again, you also might note that these places (along with machines) are effective marketing mechanisms. Keep the distinction btw advertising and marketing in mind throughout your paper (note you use “marketing” in your last sentence.”

Love that nonparametric Mann-Whitney U tests were used to compare the number of advertisements in each category by school type and school-level SES.

Reviewer #2: Thank you for the opportunity to review this article submitted to PLOS ONE, titled “Comparing the Density of Outdoor Sugar-Sweetened Beverage and Caffeinated Beverage Advertisements Near Schools by School Type and School-Level Economic Advantage”. Authors conducted a cross-sectional study of outdoor advertising of beverages in a 0.5 mile radius of middle and high schools in Texas, finding that schools with ≥60% of students eligible for FRPL had higher counts of ads for non-caffeinated sugar-sweetened beverages. While this study is timely and interesting given the increasing popularity of policy strategies to reduce SSB consumption and lack of research among youth, there are additional points of clarification that need to be addressed. Please find my comments below.

Major comments

The title, introduction, and discussion of the manuscript make reference to the density of outdoor SSB advertising, but estimates are reported as the absolute number of ads within the ½ mile radius. I do think it is more informative to report on the absolute scale (e.g. 7.5 ads) rather than the density scale (e.g. 7.5 ads/2.01 square kilometers = 3.6 ads/km^2), but authors should clarify this framing.

Abstract

1. In the methods, authors should provide a justification for why the Mann-Whitney U tests were used relative to other tests.

2. In the results, there are some inconsistencies in how results are characterized as significant. In the first sentence, authors state that lower SES schools had significantly more non-SSB caffeinated beverage ads but report a p-value of 0.07 from the Wilcoxon test. I would reframe the results by only providing the results that are statistically significant, but then discuss why analyses may have been underpowered (or other reasons why other hypothesized differences were not significant) in the discussion section of the body of the paper.

Introduction

1. What are the biological mechanisms by which increased caffeine intake may lead to the health outcomes described in the second paragraph? Are these relationships independent of the sweetener used in the beverage? Some discussion for the metabolic consequences of caffeine as it relates to health would be useful here.

2. Are there estimates of the quantity of caffeine that is consumed by school-aged children on average per day? This would help contextualize whether youth are even meeting the DGA recommendation.

3. The last two sentences of the 4th paragraph: authors state that previous research has found that neighborhood level income is associated with outdoor food/beverage advertising, but then in the next sentence state that no studies have examined densities of SSB or caffeinated beverage advertisements by neighborhood income. This seems like a contradiction to me, so some clarification is needed. In addition, there is one study by Zahid et al. (2022) that has examined differences in the density of unhealthy beverage advertising by neighborhood median household income and racial/ethnic composition: https://www.ncbi.nlm.nih.gov/pmc/articles/PMC9152783/

4. Numerous studies have also documented disparities by race/ethnicity in SSB and other types of food/beverage advertising and marketing. I think an important but missing component of this study is whether the number of ads for SSBs and caffeinated beverages differs by school-level racial/ethnic composition – especially because this demographic is provided in the descriptive statistics of the school sample.

5. Some justification is needed for the focus on school characteristics specifically, as opposed to characteristics of the neighborhood the school resides in. In this community, for example, do students generally attend schools within their neighborhoods? Or is traveling larger distances to attend a different school possible/common? Why would beverage companies make marketing decisions based on the composition of the school rather than the composition of the neighborhood?

6. The framing of the research hypotheses uses the terms “densities” and “proportion”, but authors analyzed the distributions of the absolute counts of beverage ads. Suggest rephrasing for clarity.

Methods

1. Details on study sample selection are needed. How was the sample of 34 middle schools, and 13 high schools derived? Is this sample representative of all middle/high schools in Austin? Were there any recruitment strategies needed?

2. What months/years were data collected? Since the extent and variety advertisements can change quite frequently, this information is important context to understand the results.

3. Data collection: how were location and type of advertisements classified in the field?

4. Coding: the formula for calculating the tool reliability should be reported, instead of just using the statement “using a custom formula in FileMaker”. If this is inter-rater reliability, are the estimates of reliability provided Cohen’s kappa statistic? In addition, “coding of the content of advertisements” is vague – what does this entail? Were these limited to just the coding of the types of products present? Or also the location of the ad?

5. Coding: For the estimate of percent agreement, is the standard deviation in percent units? Or should this be 2%?

6. Coding: How many advertisements included multiple beverages or products in the same ad? How were these ads coded? If an advertisement included a lineup of different Pepsi products, for example, were these coded as a single ad or multiple ads for each product?

7. Given the low number of ads broken down by school type and FRPL status, I do wonder if some of the analyses were simply underpowered. Were analyses performed that summed across the advertisement types (i.e. all SSB ads, and all caffeinated ads)?

8. School-level SES: how was the 60% threshold chosen? Is this an accepted threshold that others have used in the past?

9. Statistical analyses: School SES and type may not be the only driver of the distribution of advertisements in a neighborhood. I would consider taking an additional step in the analysis to examine how school characteristics might explain the ad distribution, independent of neighborhood level characteristics, using a regression model. It could be that neighborhood characteristics entirely drive the observed association between school SES and the number of ads in the neighborhood, so this analysis would be critical in determining whether there is something unique about the school characteristics themselves.

Results

1. Were there drivers for whether school areas had any advertisements at all? The analysis presented describes continuous exposure, but I wonder if there are aspects of the school -- such as the degree of political will of the children and families who attend the school -- that influence whether beverage companies choose to advertise their products in a given area at all.

2. Please include IQRs for the median estimates.

3. Please remove the language “approached significance”, as this is no longer standard practice for most journals and research articles.

Discussion

1. I would begin the Discussion with a summary of the findings, then move on to explanations and comparison to other literature.

2. In second paragraph, please remove language about associations “approaching” statistical significance. If there is an argument to be made about non-SSB/non-caffeinated beverages, I would just provide the medians and describe possible explanations for why the associations were not statistically significant (which the authors do, partially, in the remainder of the paragraph).

Minor comments

1. In the abstract, it would be useful if authors described when data were collected, since advertising of SSBs may experience both yearly and seasonal secular trends.

2. In the abstract methods section, please define Outdoor MEDIA.

3. In the abstract results section, I would include units for the median estimates within the parentheticals, as well as a note for which estimate corresponds to the high SES vs. low SES schools, e.g. ‘(Median for low SES schools = 7.5 ads vs. Median for high SES = 0 ads, p = 0.014)’. Estimates of dispersion (e.g. IQR for median) should also be included.

4. SSB should be defined before first use as an acronym.

6. PLOS authors have the option to publish the peer review history of their article (what does this mean?). If published, this will include your full peer review and any attached files.

Reviewer #1: No

Reviewer #2: No

<quillbot-extension-portal></quillbot-extension-portal>

---

## [Author Response · Author response to Decision Letter 0]

4 Jan 2024

Review Comments to the Author

A note to both reviewers: Thank you for your thoughtful comments. We believe they have strengthened the manuscript, we hope that you find the updated manuscript improved. In our revision process, we brought on a data scientist to the authorship team to help with the analyses needed for the revision. They identified an error in our original analyses which undercounted the number of products within advertisements. Our data were structured such that each advertisement was coded for up to four beverages – for example, if an advertisement contained a Coke, Diet Coke, Sprite, and a water, that single advertisement would contribute one product to each product type. In our original analyses, we inadvertently dropped all of the products after the first – in the example scenario, that ad would only be counted as a Coke. This resulted in an underestimate of the number of beverages truly advertised. Additionally, we have implemented a bootstrapping procedure to our Mann-Whitney U tests to improve the precision of our estimates. Given these improvements, our results have changed such that there are significant differences for all product types by school-level SES. We believe that these changes improve the rigor and reproducibility of the manuscript and hope that they are well received. 

Reviewer #1: Overall, this is a sound study and important to the field of predatory marketing of unhealthy food products. I hope you will give serious consideration to the following comments:

In the results section (and in abstract), I’d be careful of the word “significantly.” We don’t want to confuse statistical significance w. the effect size. I suppose it’s fine as it is but perhaps re-phrase to something like “schools with lower SES had a dramatically higher number of advertisements…” or something like that.

Thank you for this comment. We used the term “significant” only to refer to statistically significant differences, though we appreciate the attention to detail about confusing statistical significance with effect size. We have modified our language to specify that we are referring to statistical significance when discussing significant differences. 

You say: ““Schools with lower SES had more sugar-sweetened/caffeinated and non-sugar-sweetened/non-caffeinated beverage advertisements, though the differences were not statistically significant.” This is not a fair statement. If it’s not statistically significant, it really shouldn’t be mentioned at all. A more academically rigorous statement would be to say: “There were no statistically significant differences between schools with lower SES and higher SES in terms of sugar-sweetened/caffeinated and non-sugar-sweetened/non-caffeinated beverage advertisements.” The results are the results (and I agree that .05 p values are arbitrary). But strong papers do not emphasize (and many don’t even mention) non-significant results. Your study is sound… don’t grasp for straws when you have some perfectly solid results to report.

We agree with your suggestion to rephrase this statement. Our intent was to show that our sample size was, perhaps, underpowered to detect significant differences, given the observable disparity in median number of sugar-sweetened caffeinated beverage advertisements and non-sugar sweetened caffeinated beverages around schools with lower SES as compared to around schools with higher SES. We have since added a data scientist to the authorship team who identified an error in our original analyses, finding that a substantial portion of beverages were inadvertently dropped and thus undercounted the number of products in the advertisements. Our updated results section reflects our new findings. 

The results section does seem somewhat scant. Aren’t there more interesting results that can be reported here?

We have added some detail about the demographic composition of the included schools. Given our updated analyses, the results section has also changed to reflect our updated findings. If there are specific additions you feel would add to the interest of the manuscript, we are certainly open to feedback. 

In terms of recommendations: You might consider going beyond recommending that “that food and beverage advertising regulations should expand beyond television to include the outdoor environment around schools / protections for older youth. Remember that soda machines are themselves a marketing vehicle. So you might also mention limiting where these machines are placed.

We have added information in the discussion section with additional recommendations in response to this comment. 

In the limitations you mentioned that you didn’t consider the density of convenience stores or fast-food establishments, but again, you also might note that these places (along with machines) are effective marketing mechanisms. Keep the distinction btw advertising and marketing in mind throughout your paper (note you use “marketing” in your last sentence.”

We appreciate this comment. We have added a sentence to address this point in the discussion section and have edited the paper to better reflect the distinction between marketing and advertising. 

Love that nonparametric Mann-Whitney U tests were used to compare the number of advertisements in each category by school type and school-level SES.

Thank you for this comment! In response to other reviewer comments, we have specified that his test is non-parametric in the abstract. We have updated our analyses to include bootstrapped Mann-Whitney U tests with 1,000 repetitions to allow for better precision in our findings.

Reviewer #2: Thank you for the opportunity to review this article submitted to PLOS ONE, titled “Comparing the Density of Outdoor Sugar-Sweetened Beverage and Caffeinated Beverage Advertisements Near Schools by School Type and School-Level Economic Advantage”. Authors conducted a cross-sectional study of outdoor advertising of beverages in a 0.5 mile radius of middle and high schools in Texas, finding that schools with ≥60% of students eligible for FRPL had higher counts of ads for non-caffeinated sugar-sweetened beverages. While this study is timely and interesting given the increasing popularity of policy strategies to reduce SSB consumption and lack of research among youth, there are additional points of clarification that need to be addressed. Please find my comments below.

Major comments

The title, introduction, and discussion of the manuscript make reference to the density of outdoor SSB advertising, but estimates are reported as the absolute number of ads within the ½ mile radius. I do think it is more informative to report on the absolute scale (e.g. 7.5 ads) rather than the density scale (e.g. 7.5 ads/2.01 square kilometers = 3.6 ads/km^2), but authors should clarify this framing.

Thank you for this suggestion – we have modified our language throughout the manuscript to more accurately describe our findings. We have opted to use “number” as opposed to “density.”

Abstract

1. In the methods, authors should provide a justification for why the Mann-Whitney U tests were used relative to other tests.

We used Mann-Whitney U tests because the data did not meet normal distribution assumptions. For brevity in the abstract, we added the term “non-parametric Mann-Whitney U tests” to contextualize this decision. 

2. In the results, there are some inconsistencies in how results are characterized as significant. In the first sentence, authors state that lower SES schools had significantly more non-SSB caffeinated beverage ads but report a p-value of 0.07 from the Wilcoxon test. I would reframe the results by only providing the results that are statistically significant, but then discuss why analyses may have been underpowered (or other reasons why other hypothesized differences were not significant) in the discussion section of the body of the paper.

Thank you for this comment – the p-value presented should have been 0.007. As noted above, we implemented additional improvements to our analyses and as a result our findings have changed. This section now reflects our updated findings. 

Introduction

1. What are the biological mechanisms by which increased caffeine intake may lead to the health outcomes described in the second paragraph? Are these relationships independent of the sweetener used in the beverage? Some discussion for the metabolic consequences of caffeine as it relates to health would be useful here.

We have included a brief discussion of caffeine as a psychostimulant that acts on the nervous system, but for brevity, we did not go into detail on the mechanisms between caffeine intake and each health consequence described. This is outside of our scope of expertise, but have included citations to refer readers to literature on the topic. While there are gaps in the literature specifically examining if these relationships are independent of sweeteners, caffeine’s association with obesity in children is likely due to its presence in sugary drinks. We have included this information in this manuscript. 

2. Are there estimates of the quantity of caffeine that is consumed by school-aged children on average per day? This would help contextualize whether youth are even meeting the DGA recommendation.

We have added a sentence to the introduction describing these estimates. Thank you for this suggestion.

3. The last two sentences of the 4th paragraph: authors state that previous research has found that neighborhood level income is associated with outdoor food/beverage advertising, but then in the next sentence state that no studies have examined densities of SSB or caffeinated beverage advertisements by neighborhood income. This seems like a contradiction to me, so some clarification is needed. In addition, there is one study by Zahid et al. (2022) that has examined differences in the density of unhealthy beverage advertising by neighborhood median household income and racial/ethnic composition: https://www.ncbi.nlm.nih.gov/pmc/articles/PMC9152783/

In this paragraph, we intended to highlight that no research to date has specifically examined if the types of beverages advertised (i.e., sugar-sweetened, non-sugar-sweetened, caffeinated, non-caffeinated) in outdoor environments vary by neighborhood makeup, which we believe is still the case. Thank you for sharing the Zahid et al article, we appreciate that this study did look at beverage advertisements specifically. However, the Zahid et al study focuses on sweetened beverages and did not consider beverage caffeine content. We have modified this paragraph to clarify our intent and to include findings from the referenced study. 

4. Numerous studies have also documented disparities by race/ethnicity in SSB and other types of food/beverage advertising and marketing. I think an important but missing component of this study is whether the number of ads for SSBs and caffeinated beverages differs by school-level racial/ethnic composition – especially because this demographic is provided in the descriptive statistics of the school sample.

We considered examining if beverage advertisements varied by the racial/ethnic composition of the included schools. However, the racial/ethnic composition of the included schools is colinear with the school-level FRPL – that is, schools with a high percentage of non-white students also had a high percentage of students qualifying for FRPL. We have added a sentence noting this decision in the statistical analysis section. 

5. Some justification is needed for the focus on school characteristics specifically, as opposed to characteristics of the neighborhood the school resides in. In this community, for example, do students generally attend schools within their neighborhoods? Or is traveling larger distances to attend a different school possible/common? Why would beverage companies make marketing decisions based on the composition of the school rather than the composition of the neighborhood?

We appreciate the opportunity to address this question. School-level characteristics were chosen for two primary reasons: first, the purpose of the study is to examine the outdoor advertising environments near schools – thus, the students attending these schools are the primary population of concern in our study. Previous research has found that school-level SES is an adequate proxy for household income, thus providing insight to the income level of community the school serves. Secondly, there are no publicly available data, to our knowledge, that map onto the ½ mile radius buffer zone around schools measured in this study and contain relevant community-level characteristics. As stated in the paper, the ½ mile radius was chosen because it reflects a similar radius used in other studies examining the school environment, and because it is a reasonable distance for students to walk (e.g., to get a snack after school). We do not have data on the distance students travel to school in Austin, specifically, though data from the National Center for Educational Statistics suggests that the majority of students in the United States attend public schools within their district (72.8%). It is possible that students travel outside of their communities to attend school, but it is also reasonable that even if they do not attend their neighborhood school for their education, they are likely pass through the area regularly or use the facilities on the before school, after school, or on the weekends. Thus, we can reasonably assume that the school-level characteristics used in this study are representative of the communities they serve. While we cannot say for certain whether advertisers make conscious decisions based on school-level characteristics, they certainly do make advertising decisions based on community-level characteristics. The reality is that the students attending these schools are frequently exposed to the advertisements. We hope that this provides adequate justification for our use of school-level demographics. We have added a brief paragraph to the manuscript to address this comment. 

6. The framing of the research hypotheses uses the terms “densities” and “proportion”, but authors analyzed the distributions of the absolute counts of beverage ads. Suggest rephrasing for clarity.

We have modified the language of our hypotheses to reflect this suggestion. 

Methods

1. Details on study sample selection are needed. How was the sample of 34 middle schools, and 13 high schools derived? Is this sample representative of all middle/high schools in Austin? Were there any recruitment strategies needed?

We included all 32 middle and high schools in the Austin Independent School District (AISD), as well as 15 other middle schools from nearby areas. Initially, our study was designed to integrate data with the Coordinated Approach to Child Health (CATCH) Middle School Study (a sample of middle schools in AISD and from surrounding school districts), but ultimately did not. However, when we began the study, we decided to document all CATCH schools as well as all middle and high schools in AISD. We have added additional detail to the methods section. 

2. What months/years were data collected? Since the extent and variety advertisements can change quite frequently, this information is important context to understand the results.

Data were collected in 2012 – we apologize for our oversight in not including this in the original manuscript. As you mention, advertising practices can change frequently and with seasonality, we discovered this during data collection (some advertisements changed overnight!) However, while these data are older, there is no evidence that outdoor advertising practices have changed significantly. Additionally, no policies have been implemented that would affect beverage industry practices in Austin. Research using data from the Outdoor MEDIA study was published this year, and we believe that insights from this dataset can still contribute t

---

## [Decision Letter · Decision Letter 1]

28 Feb 2024

PONE-D-23-10045R1Comparing the Number of Outdoor Sugar-Sweetened Beverage and Caffeinated Beverage Advertisements Near Schools by School Type and School-Level Economic AdvantagePLOS ONE

Dear Dr. Pasch,

Thank you for submitting your manuscript to PLOS ONE. After careful consideration, we feel that it has merit but does not fully meet PLOS ONE’s publication criteria as it currently stands. Therefore, we invite you to submit a revised version of the manuscript that addresses the points raised during the review process.

We look forward to receiving your revised manuscript.

Kind regards,

Habiba I. Ali, PhD, RD, CDE

Academic Editor

PLOS ONE

Journal Requirements:

Additional Editor Comments:

Please address the reviewers' comments.

**Comments to the Author**

Reviewer #1: All comments have been addressed

Reviewer #2: All comments have been addressed

2. Is the manuscript technically sound, and do the data support the conclusions?

Reviewer #1: Yes

Reviewer #2: Yes

3. Has the statistical analysis been performed appropriately and rigorously? 

Reviewer #1: Yes

Reviewer #2: Yes

4. Have the authors made all data underlying the findings in their manuscript fully available?

Reviewer #1: Yes

Reviewer #2: Yes

5. Is the manuscript presented in an intelligible fashion and written in standard English?

Reviewer #1: Yes

Reviewer #2: Yes

6. Review Comments to the Author

Reviewer #1: Nice job responding to our comments. The only suggestion I have is to update your background - or at least this sentence: "As of 2014, nearly two thirds of children in the United States consumed at least one SSB per day." Since 2014 is now ten years ago it seems odd to include data so old. This is still a problem in 2024 so I request that you add at least one sentence that demonstrates this (i.e. data from 2019 at the earliest, but ideally from 2022 or '23 perhaps?)

Reviewer #2: Thank you for your close attention to the comments from reviewers, and for being transparent and upfront about the discovery of a coding error that altered your findings. I have no additional major comments. My one additional request is that you include a reference for the bootstrapping procedure used. Apart from that, I look forward to reading the final version of the manuscript.

7. PLOS authors have the option to publish the peer review history of their article (what does this mean?). If published, this will include your full peer review and any attached files.

Reviewer #1: No

Reviewer #2: No

---

## [Author Response · Author response to Decision Letter 1]

18 Mar 2024

Response to Reviewers

Reviewer #1: Nice job responding to our comments. The only suggestion I have is to update your background - or at least this sentence: "As of 2014, nearly two thirds of children in the United States consumed at least one SSB per day." Since 2014 is now ten years ago it seems odd to include data so old. This is still a problem in 2024 so I request that you add at least one sentence that demonstrates this (i.e. data from 2019 at the earliest, but ideally from 2022 or '23 perhaps?)

Thank you. We have revised this sentence to report the most recent data for the United States that were nationally representative, however, the most recent data were for 2018. If there is a specific paper that the reviewer is aware of that reports on the prevalence of sugar-sweetened beverage consumption for the US that is more recent we are happy to revise.

Reviewer #2: Thank you for your close attention to the comments from reviewers, and for being transparent and upfront about the discovery of a coding error that altered your findings. I have no additional major comments. My one additional request is that you include a reference for the bootstrapping procedure used. Apart from that, I look forward to reading the final version of the manuscript.

Thank you. We have added a reference for the bootstrapping procedure.

---

## [Editor Report · Decision Letter 2]

27 Mar 2024

Comparing the Number of Outdoor Sugar-Sweetened Beverage and Caffeinated Beverage Advertisements Near Schools by School Type and School-Level Economic Advantage

PONE-D-23-10045R2

Dear Dr. Pasch,

We’re pleased to inform you that your manuscript has been judged scientifically suitable for publication and will be formally accepted for publication once it meets all outstanding technical requirements.

Kind regards,

Habiba I. Ali, PhD, RD, CDE

Academic Editor

PLOS ONE

Additional Editor Comments (optional):

The authors have addressed the comments of the reviewers.
---

## [Editor Report · Acceptance letter]

14 May 2024

PONE-D-23-10045R2 

PLOS ONE

Dear Dr. Pasch, 

I'm pleased to inform you that your manuscript has been deemed suitable for publication in PLOS ONE. Congratulations! Your manuscript is now being handed over to our production team.

Kind regards, 

on behalf of

Dr. Habiba I. Ali 

Academic Editor

PLOS ONE